# Computed Tomography-Based Sarcopenia and Pancreatic Cancer Survival—A Comprehensive Meta-Analysis Exploring the Influence of Definition Criteria, Prevalence, and Treatment Intention

**DOI:** 10.3390/cancers17040607

**Published:** 2025-02-11

**Authors:** Antonio Jesús Láinez Ramos-Bossini, Antonio Gámez Martínez, David Luengo Gómez, Francisco Valverde-López, Antonio Jesús Morillo Gil, Encarnación González Flores, Ángela Salmerón Ruiz, Paula María Jiménez Gutiérrez, Consolación Melguizo, José Prados

**Affiliations:** 1Department of Radiology, Hospital Universitario Virgen de las Nieves, 18014 Granada, Spain; antonio.gam.mar@gmail.com (A.G.M.); davidluengog@gmail.com (D.L.G.); antoniojesusmorillo@gmail.com (A.J.M.G.); salmeronruiz@gmail.com (Á.S.R.); 2Advanced Medical Imaging Group (TeCe-22), Instituto de Investigación Biosanitaria de Granada (ibs.GRANADA), 18016 Granada, Spain; 3Department of Human Anatomy and Embryology, Faculty of Medicine, University of Granada, 18071 Granada, Spain; cmelguizo@ugr.es (C.M.); jcprados@ugr.es (J.P.); 4Department of Gastroenterology and Hepatology, Hospital Universitario Virgen de las Nieves, 18014 Granada, Spain; fcovalverde89@gmail.com; 5Department of Medical Oncology, Hospital Universitario Virgen de las Nieves, 18014 Granada, Spain; encarnagonzalezflores@gmail.com; 6Department of Anesthesiology, Hospital Universitario Virgen de las Nieves, 18014 Granada, Spain; apaulajimenezg@gmail.com; 7Institute of Biopathology and Regenerative Medicine (IBIMER), University of Granada, 18100 Granada, Spain; 8Center of Biomedical Research (CIBM), University of Granada, 18100 Granada, Spain

**Keywords:** sarcopenia, pancreatic cancer, overall survival, progression-free survival, meta-analysis

## Abstract

Sarcopenia, a condition characterized by the loss of skeletal muscle mass, is increasingly recognized as a predictor of poor outcomes in pancreatic cancer (PC). This meta-analysis investigates the impact of sarcopenia, as assessed by computed tomography (CT), on survival outcomes in PC patients. We analyzed data from multiple studies to determine how sarcopenia affects cancer-related outcomes, particularly overall survival (OS) and progression-free survival (PFS). Our findings show that sarcopenia significantly worsens both OS and PFS. This negative impact is more pronounced in patients undergoing curative treatments and when stricter CT-based cutoff values are applied. These results highlight the importance of routinely evaluating sarcopenia in clinical settings, as early identification may guide treatment decisions and improve patient outcomes. Future research should explore strategies to manage sarcopenia and further standardize its measurement.

## 1. Introduction

Pancreatic cancer (PC) represents one of the most lethal malignancies, with a 5-year survival rate of less than 5–10%, despite advances in surgical techniques and systemic therapies [1,2]. Its poor prognosis can be attributed to a variety of factors, including late-stage diagnosis, aggressive tumor biology, and limited efficacy of available treatments [3]. Consequently, identifying prognostic factors that can inform clinical decision-making is essential to improve the management and outcomes of PC. In this context, sarcopenia has recently emerged as a key determinant of patient prognosis, particularly in the context of cancer cachexia [4].

Sarcopenia, defined as the progressive and generalized loss of skeletal muscle mass and strength [5], was initially recognized as an age-related condition. However, it is now increasingly conceived as a pathological state that can affect individuals of all ages, especially those with chronic diseases, including cancer [6]. The association between sarcopenia and poor outcomes in PC patients has gained substantial attention in recent years, with numerous studies suggesting that sarcopenia may be an independent predictor of worse overall survival (OS) and progression-free survival (PFS) [7]. The mechanisms linking sarcopenia to adverse oncological outcomes are likely related to the systemic effects of chronic inflammation, metabolic dysregulation, and reduced tolerance to anticancer therapies [8]. However, the exact pathophysiology is still poorly understood, which has motivated increasing research interest on the topic.

In PC, the systemic inflammatory response associated with the disease contributes to the early development of sarcopenia [9]. Tumor-induced alterations in metabolism, including increased energy expenditure and altered protein synthesis, exacerbate the loss of muscle mass [10]. Sarcopenia may also reflect underlying frailty in these patients, reducing their response to the physiological stress of both PC and its treatments. This frailty may manifest clinically as increased susceptibility to postoperative complications, reduced physical performance, and diminished ability to recover from surgery or systemic therapy [11,12].

A critical challenge in studying sarcopenia in cancer patients is the accurate assessment of muscle mass. Traditional methods such as bioelectrical impedance analysis and dual-energy X-ray absorptiometry have been employed in the past, but their accessibility and precision are limited in the oncology setting [13]. In contrast, computed tomography (CT) scans, which are routinely performed in cancer patients for diagnostic and staging purposes, have emerged as a valuable tool for the objective measurement of skeletal muscle mass [14]. This imaging modality allows for the simultaneous evaluation of sarcopenia and tumor burden, facilitating the incorporation of muscle mass measurements into routine clinical practice without the need for additional testing [15].

CT-based assessment of sarcopenia typically involves the quantification of muscle cross-sectional area at the level of the third lumbar vertebra (L3), which has been shown to correlate with total body skeletal muscle mass [16]. However, a number of methods and quantitative thresholds to define sarcopenia have been proposed in the literature, including but not limited to skeletal muscle index (SMI) and total psoas area (TPA) and volume (TPV). Unfortunately, no consensus on the most suitable one exists to date [17]. Such measurements have been used to investigate the impact of sarcopenia on PC outcomes (e.g., postoperative morbidity, chemotherapy toxicity, survival) [18].

To date, several studies have demonstrated that sarcopenia is associated with worse OS and PFS in PC patients [19], and it has also been linked to increased rates of postoperative complications in patients undergoing pancreaticoduodenectomy, which is the standard surgical treatment for resectable PC [20]. Moreover, in patients with locally advanced or metastatic disease, sarcopenia has been associated with reduced chemotherapy tolerance, leading to dose reductions or treatment delays that negatively impact survival [21]. These findings suggest that sarcopenia could serve as a biomarker for treatment stratification, guiding decisions on dose adjustments or the use of supportive therapies to mitigate treatment-related toxicities [22].

However, despite the accumulating evidence supporting the association between sarcopenia and poor outcomes in PC, there is significant variability in the reported outcomes across studies due to differences in the methods used to assess sarcopenia, the inclusion criteria of the studied populations, and the use of different CT-based thresholds for defining sarcopenia, among other factors [23]. Given the growing interest in the role of sarcopenia as a prognostic factor in PC, there is a need for a comprehensive synthesis of the available evidence.

The objective of this study is to conduct a meta-analysis of observational studies to evaluate the influence of CT-defined sarcopenia on OS and PFS in patients with PC. We previously reported meta-analytic data on the variability of sarcopenia prevalence based on the CT-based method and thresholds used in its definition [24]. In the present article, we aim to provide a comprehensive assessment of the prognostic value of sarcopenia in this patient population, examining the role of potential confounders such as the imaging methods and cutoff values used to define sarcopenia, which may have important implications for clinical practice and future research.

## 2. Materials and Methods

### 2.1. Eligibility Criteria

The selection criteria included observational studies of patients with histologically confirmed PC, regardless of treatment intention, reporting the prevalence of pre-treatment sarcopenia determined by CT as well as survival outcomes, particularly OS or PFS. The Preferred Reporting Items for Systematic Reviews and Meta-Analyses (PRISMA) [25] guidelines were followed in the design and writing of the study (see Appendix A for the PRISMA checklist). This review was not registered in publicly available registers, but an internal protocol in agreement with the PRISMA guidelines (Appendix A) was followed. The exclusion criteria, as in our previous meta-analysis, included studies reporting no mortality-related outcomes, articles published in languages other than English, studies with incomplete data on sarcopenia prevalence, or studies and publications different from original research articles reporting data from observational studies (e.g., reviews, case reports and series, conference proceedings, letters to the editor).

### 2.2. Information Sources and Search Strategy

The information sources and search strategy used are analogous to those described in our previous work [24]. In brief, two authors searched the PubMed, Web of Science, and EMBASE databases to identify original studies published from database inception until 26 April 2024. Different search strategies were carried out, and a final consistent equation was constructed (Appendix A). To increase the sensitivity of the search, references of all fully read articles were also examined. No date or language restrictions were established.

All titles and abstracts of interest were screened, and those which did not meet the eligibility criteria were excluded. Subsequently, the screened studies were fully read to assess whether they met all eligibility criteria. Figure 1 shows the flow diagram of the study.

### 2.3. Measured Variables and Subgroup Analyses

Data were collected regarding study characteristics, patient population, sarcopenia measurement, and cancer-related characteristics, including treatment intention (i.e., curative vs. palliative). The primary outcomes were OS and PFS, which were analyzed separately based on whether original studies reported univariate (i.e., crude HRs—cHRs) or multivariate (i.e., adjusted HRs—aHRs) analysis results. In addition, we performed subgroup analyses based on the following variables:Method used to calculate sarcopenia (SMI or analogous measurement vs. other measurement such as TPA or TPV).For studies measuring sarcopenia using SMI, we analyzed between-group differences to compare studies defining sarcopenia below or over the threshold of 50 cm^2^/m^2^. As in our previous meta-analysis [24], when a study reported the prevalence of sarcopenia using different cutoffs, the sample was split or duplicated accordingly and independently analyzed.Oncological context in terms of patient management, namely palliative (non-resectable or metastatic cancer) or curative (managed with surgery with or without chemo/radiotherapy).

Studies not reporting any of these variables were excluded from the corresponding subgroup analysis.

### 2.4. Data Extraction and Quality Assessment

Two authors (D.LG. and F.V.L.) independently extracted the data from the selected articles, and a senior author (J.P.) reviewed the data and solved any discrepancies. If there were several definitions for sarcopenia, we included the one which appeared significant for survival analyses in the study. All data were stored using a spreadsheet designed for such purpose. The quality assessment of the included studies was performed using the Newcastle–Ottawa scale (NOS) [26], which can be consulted in our previous work [24].

### 2.5. Statistical Analysis

We applied the inverse-variance weighting method with a random-effects model, using the Hartung–Knapp (HK) adjustment to calculate the confidence intervals for the combined effect. This adjustment provides more conservative estimates of the standard error and confidence intervals, especially in situations with high heterogeneity among studies. Heterogeneity among studies was assessed using the I^2^ statistic, with cutoff values set at I^2^ < 40% as non-relevant, 40% < I^2^ < 75% as moderate, and I^2^ > 75% as high, as in previous meta-analyses [24,27,28]. We also assessed the τ^2^ statistic, which provides a quantitative estimate of the between-study variance [29].

Additionally, we calculated the prediction interval (PI) for the combined effect, which estimates the range within which the true effect of a new, similar study is expected to fall. Unlike the confidence interval for the average effect, the prediction interval accounts for both the uncertainty of the average effect and the variability among studies [30].

To explore the robustness of our results, we conducted leave-one-out sensitivity analyses. Finally, publication bias was assessed using funnel plots and Egger’s test for funnel plot asymmetry.

*p*-values < 0.05 were considered statistically significant. All statistical analyses were carried out with software R (version 4.3.2 for Windows) [31] using the ‘meta’ package [32].

## 3. Results

### 3.1. Characteristics of the Included Studies

A total of 48 studies were included in the meta-analysis [7,23,33,34,35,36,37,38,39,40,41,42,43,44,45,46,47,48,49,50,51,52,53,54,55,56,57,58,59,60,61,62,63,64,65,66,67,68,69,70,71,72,73,74,75,76,77,78]. As in our previous work, two studies provided separated measures for their patient cohorts and were thus split into two different studies for analyses [73,74]. Therefore, 50 studies were considered in data analysis, encompassing data from 9063 patients in the original cohorts (45% women, sample sizes ranging from 41 to 763 patients).

Most studies (43/50, 86%) applied SMI or an analogous measurement to estimate sarcopenia, while seven (14%) studies applied other measurements. Details regarding the characteristics of the included studies and sarcopenia-related measurements are shown in Table 1 (further details can be consulted in our previous work [24]). Notably, sarcopenia was defined with a cutoff value < 50 cm^2^/m^2^ for males in 26 out of 38 (68.4%) studies reporting aHR, in 21 out of 29 studies (72.4%) reporting cHR values for OS, in 12 out of 15 (80%) reporting aHR values for PFS, and in 9 out of 11 (81.8%) studies reporting cHR values for PFS.

### 3.2. Sarcopenia as a Risk Factor for Overall Survival

#### 3.2.1. Meta-Analysis of Overall Survival Based on Univariate Regression Analyses

A total of 45 studies encompassing 8389 patients reported cHR values for OS. The pooled HR for OS was 1.58 (95% CI, 1.38–1.82), indicating that sarcopenia was significantly associated with worse OS. Significant heterogeneity was observed across the studies (I^2^ = 85%, τ^2^ = 0.15, *p* < 0.01). The cHR values reported by individual studies ranged from 0.81 (95% CI, 0.52–1.25) to 6.90 (95% CI, 1.68–28.40). The PI ranged from 0.71 to 3.54, suggesting that while sarcopenia is generally associated with worse OS, the magnitude of this effect may vary across different settings. Figure 2 shows the forest plot of the meta-analysis of OS based on univariate regression analyses.

#### 3.2.2. Meta-Analysis of Overall Survival Based on Multivariate Regression Analyses

A total of 36 studies encompassing 7619 patients reported aHR values for OS. The pooled HR for OS was 1.68 (95% CI, 1.48–1.91), indicating that sarcopenia was significantly associated with worse OS. Significant heterogeneity was observed across the studies (I^2^ = 82%, τ^2^ = 0.09, *p* < 0.01). The cHR values reported by individual studies ranged from 0.94 (95% CI, 0.87–1.01) to 5.67 (95% CI, 3.58–8.98). The PI ranged from 0.90 to 3.13, suggesting that the magnitude of the association may vary across different settings. Figure 3 shows the forest plot of the meta-analysis of OS based on multivariate regression analyses.

### 3.3. Sarcopenia as a Risk Factor for Progression-Free Survival

#### 3.3.1. Meta-Analysis of Progression-Free Survival Based on Univariate Regression Analyses

A total of 19 studies, including 2973 patients, reported cHR values for PFS. The pooled HR for PFS was 1.39 (95% CI, 1.16–1.66), indicating that sarcopenia was significantly associated with worse PFS. Substantial heterogeneity was observed across the studies (I^2^ = 78%, τ^2^ = 0.09, *p* < 0.01). The cHR values reported by individual studies ranged from 0.38 (95% CI, 0.13–1.11) to 2.59 (95% CI, 1.79–3.74). The PI ranged from 0.73 to 2.65, suggesting that the extent of this effect may vary depending on the specific clinical context. Figure 4 presents the forest plot of the meta-analysis of PFS based on univariate regression analyses.

#### 3.3.2. Meta-Analysis of Progression-Free Survival Based on Multivariate Regression Analyses

A total of 15 studies, including 2635 patients, reported aHR values for PFS. The pooled HR for PFS was 1.55 (95% CI, 1.29–1.86), indicating that sarcopenia was significantly associated with worse PFS in multivariate analyses. There was moderate heterogeneity across the studies (I^2^ = 67%, τ^2^ = 0.04, *p* < 0.01). The reported aHR values ranged from 0.37 (95% CI, 0.12–1.14) to 3.44 (95% CI, 1.57–7.54). The PI ranged from 0.97 to 2.46, suggesting that the magnitude of this effect could vary in different clinical scenarios. Figure 5 presents the forest plot of the meta-analysis of PFS based on multivariate regression analyses.

### 3.4. Subgroup Analyses

#### 3.4.1. Subgroup Analysis Based on the Method Used to Estimate Sarcopenia

-Overall survival (univariate analyses, cHR): The first subgroup analysis explored the impact of the method used to define sarcopenia on OS based on univariate regression analyses’ reported outcomes. The 38 studies that employed SMI as the method to define sarcopenia showed a pooled cHR of 1.48 (95% CI, 1.28; 1.72), with high heterogeneity (I^2^ = 82%, τ^2^ = 0.12, *p* < 0.01). On the other hand, the seven studies using other methods to define sarcopenia reported a pooled cHR of 2.17 (1.35; 3.48), also showing substantial heterogeneity (I^2^ = 83%, τ^2^ = 0.22, *p* < 0.01). The test for subgroup differences showed a trend toward statistical significance (*p* = 0.07), thus no strong evidence of differential effects based on the method used to define sarcopenia was observed (Appendix A).-Overall survival (multivariate analyses, aHR): In the subgroup analysis for OS based on multivariate regression analyses’ reported outcomes, the pooled aHR for the 29 studies defining sarcopenia using SMI was 1.59 (1.40; 1.80), with substantial heterogeneity (I^2^ = 79%, τ^2^ = 0.06, *p* < 0.01). Studies using other methods yielded a pooled HR of 1.98 (1.20; 3.26), also demonstrating significant heterogeneity (I^2^ = 83%, τ^2^ = 0.24, *p* < 0.01). The test for subgroup differences did not reveal significant differences between the subgroups (*p* = 0.29), indicating no clear difference in the effect of sarcopenia on survival based on the method used for its definition (Appendix A).-Progression-free survival (univariate analyses, cHR): For PFS based on univariate regression analyses’ reported outcomes, the 14 studies using SMI to define sarcopenia demonstrated a pooled cHR of 1.33 (1.11; 1.60), with moderate heterogeneity (I^2^ = 74%, τ^2^ = 0.07, *p* < 0.01). Studies using other methods for defining sarcopenia showed a pooled HR of 1.44 (0.65; 3.19), with high heterogeneity (I^2^ = 80%, τ^2^ = 0.25, *p* < 0.01). The test for subgroup differences was not statistically significant (*p* = 0.80), indicating no significant differences between the methods used to define sarcopenia in relation to PFS (Appendix A).-Progression-free survival (multivariate analyses, aHR): In the final subgroup analysis for PFS based on multivariate regression analyses, the pooled HR for the 10 studies employing SMI was 1.53 (1.30; 1.81), with moderate heterogeneity (I^2^ = 62%, τ^2^ = 0.02, *p* < 0.01). In contrast, studies utilizing other methods to define sarcopenia reported a pooled HR of 1.40 (0.65; 3.03), showing high heterogeneity (I^2^ = 78%, τ^2^ = 0.23, *p* < 0.01). The test for subgroup differences did not reveal significant differences (*p* = 0.75), suggesting that the method used to define sarcopenia did not significantly alter the association with PFS (Appendix A).

#### 3.4.2. Subgroup Analysis Based on the Cutoff Used in SMI

In this subgroup analysis, the studies were divided based on the cutoff value used for the skeletal muscle index (SMI) to define sarcopenia. Studies were categorized as using a cutoff of either <50 cm^2^/m^2^ or >50 cm^2^/m^2^.

-Overall survival (univariate analyses, cHR): For OS based on univariate analyses’ reported outcomes, the pooled cHR for the 26 studies using an SMI cutoff of <50 cm^2^/m^2^ was 1.63 (95% CI: 1.34, 1.98), whereas for studies using an SMI cutoff of >50 cm^2^/m^2^, the pooled HR was 1.23 (95% CI: 1.02, 1.48). The heterogeneity in the subgroup using <50 cm^2^/m^2^ was high (I^2^ = 85%, τ^2^ = 0.16, *p* < 0.01), while heterogeneity for the >50 cm^2^/m^2^ subgroup was lower (I^2^ = 65%, τ^2^ = 0.05, *p* < 0.01). The test for subgroup differences reached statistical significance, indicating that the cutoff used to define sarcopenia altered the association with OS (*p* = 0.03) (Appendix A).-Overall survival (multivariate analyses, aHR): For OS based on multivariate analysis reported outcomes, the pooled aHR for the 21 studies with an SMI cutoff of <50 cm^2^/m^2^ was 1.70 (95% CI: 1.46, 1.98), whereas the HR for studies using >50 cm^2^/m^2^ was 1.32 (95% CI: 1.05, 1.66). Similar to the univariate analyses, the heterogeneity was higher for the <50 cm^2^/m^2^ subgroup (I^2^ = 83%, τ^2^ = 0.06, *p* < 0.01) compared to the >50 cm^2^/m^2^ subgroup (I^2^ = 49%, τ^2^ = 0.03, *p* < 0.01). The difference between subgroups also showed statistical significance (*p* = 0.04) (Appendix A).-Progression free survival (univariate analyses, cHR): Regarding PFS based on univariate analysis reported outcomes, the pooled cHR for the <50 cm^2^/m^2^ subgroup (12 studies) was 1.38 (95% CI: 1.14, 1.67), with moderate heterogeneity (I^2^ = 55%, τ^2^ = 0.05, *p* < 0.01). For the >50 cm^2^/m^2^ subgroup, the pooled cHR was 1.15 (95% CI: 0.51, 2.59), although the heterogeneity was significantly higher (I^2^ = 85%, τ^2^ = 0.08, *p* < 0.01). The test for subgroup differences showed no statistically significant differences (*p* = 0.38) (Appendix A).-Progression-free survival (multivariate analyses, aHR): Finally, for PFS based on multivariate analysis reported outcomes, the pooled HR for the <50 cm^2^/m^2^ subgroup (nine studies) was 1.54 (95% CI: 1.30, 1.81), while the HR for the >50 cm^2^/m^2^ subgroup was 1.32 (95% CI: 0.14, 12.49). Heterogeneity for the <50 cm^2^/m^2^ group was low (I^2^ = 36%, τ^2^ = 0.01, *p* < 0.01) compared to that of the >50 cm^2^/m^2^ subgroup, which was moderate (I^2^ = 47%, τ^2^ = 0.04, *p* < 0.01). There were no significant subgroup differences (*p* = 0.42) (Appendix A).

#### 3.4.3. Subgroup Analysis Based on the Prevalence of Sarcopenia Found in Each Study

The subgroup analysis aimed at comparing the impact of sarcopenia prevalence on OS and PFS outcomes based on studies reporting sarcopenia prevalence below and above 50%.

-Overall survival (univariate analyses, cHR): The 20 studies reporting a prevalence of sarcopenia ≥ 50% yielded a pooled cHR of 1.40 [95% CI, 1.17–1.67] with moderate heterogeneity (I^2^ = 79%, τ^2^ = 0.09, *p* < 0.01). On the other hand, studies with a prevalence < 50% demonstrated a pooled cHR of 1.72 (95% CI, 1.39–2.14) with similarly high heterogeneity (I^2^ = 79%, τ^2^ = 0.20, *p* < 0.01). The test for subgroup differences was not statistically significant (*p* = 0.12), indicating no significant differences between the two groups (Appendix A).-Overall survival (multivariate analyses, aHR): The pooled aHR for the 14 studies with sarcopenia prevalence ≥ 50% was 1.59 (95% CI, 1.29–1.96), with substantial heterogeneity (I^2^ = 80%, τ^2^ = 0.08, *p* < 0.01). For studies with prevalence < 50%, the pooled aHR was 1.73 (95% CI, 1.45–2.06), and heterogeneity was moderate (I^2^ = 69%, τ^2^ = 0.10, *p* < 0.01). There were no significant between-group differences (*p* = 0.51), suggesting that the prevalence of sarcopenia did not significantly influence OS outcomes in multivariate analyses (Appendix A).-Progression-free survival (univariate analyses, cHR): The 11 studies with sarcopenia prevalence ≥ 50% showed a pooled cHR of 1.56 (95% CI, 1.18–2.06), with moderate heterogeneity (I^2^ = 65%, τ^2^ = 0.08, *p* < 0.01). In contrast, studies reporting sarcopenia prevalence < 50% had a pooled cHR of 1.19 (95% CI, 0.94–1.51), with slightly higher heterogeneity (I^2^ = 71%, τ^2^ = 0.05, *p* < 0.01). The test for subgroup differences approached statistical significance (*p* = 0.09), suggesting but not confirming that the prevalence of sarcopenia might have some influence on PFS outcomes in univariate analyses (Appendix A).-Progression-free survival (multivariate analyses, aHR): The 10 studies with sarcopenia prevalence ≥ 50% yielded a pooled aHR of 1.61 (95% CI, 1.22–2.14), with moderate heterogeneity (I^2^ = 63%, τ^2^ = 0.06, *p* < 0.01). The pooled aHR for studies with prevalence < 50% was 1.41 (95% CI, 1.08–1.84), with lower heterogeneity (I^2^ = 38%, τ^2^ = 0.02, *p* < 0.01). The test for subgroup differences was not statistically significant (*p* = 0.39), suggesting no impact of sarcopenia prevalence on PFS outcomes in multivariate analyses (Appendix A).

#### 3.4.4. Subgroup Analysis Based on Treatment Intention in Each Study

The subgroup analysis considering treatment intention (curative vs. palliative) revealed differences in the HR values for OS and PFS.

-Overall survival (univariate analyses, cHR): The pooled cHR for the 25 studies in the curative setting was 1.75 (95% CI, 1.44–2.12), with significant heterogeneity (I^2^ = 80%, τ^2^ = 0.15, *p* < 0.01). In contrast, the palliative subgroup yielded a cHR of 1.40 (95% CI, 1.13–1.73) with similar heterogeneity (I^2^ = 81%, τ^2^ = 0.12, *p* < 0.01). Although the point estimates between subgroups were different, the test for subgroup differences did not reach statistical significance (*p* = 0.10) (Appendix A).-Overall survival (multivariate analyses, aHR): The curative setting subgroup (23 studies) showed a pooled aHR of 1.74 (95% CI, 1.46–2.08), with significant heterogeneity (I^2^ = 72%, τ^2^ = 0.11, *p* < 0.01). In the palliative subgroup, the pooled aHR was 1.54 (95% CI, 1.25–1.90), also with notable heterogeneity (I^2^ = 86%, τ^2^ = 0.07, *p* < 0.01). The test for subgroup differences was not significant (*p* = 0.33) (Appendix A).-Progression-free survival (univariate analyses, cHR): The curative setting subgroup (eight studies) resulted in a pooled cHR of 1.53 (95% CI, 1.23–1.90), with moderate heterogeneity (I^2^ = 60%, τ^2^ = 0.04, *p* < 0.01), while the palliative setting subgroup had a pooled cHR of 1.09 (95% CI, 0.86–1.38) with similar heterogeneity (I^2^ = 58%, τ^2^ = 0.03, *p* < 0.01). The difference between the two settings was statistically significant (*p* = 0.01) (Appendix A).-Progression-free survival (multivariate analyses, aHR): The curative subgroup (11 studies) produced a pooled aHR of 1.63 (95% CI, 1.28–2.08) with low heterogeneity (I^2^ = 73%, τ^2^ = 0.05, *p* < 0.01). The palliative subgroup, on the other hand, showed a pooled aHR of 1.35 (95% CI, 0.94–1.94) with non-significant heterogeneity (I^2^ = 30%, τ^2^ = 0.01, *p* = 0.23). The test for subgroup differences indicated no significant difference between the two settings (*p* = 0.23) (Appendix A).

### 3.5. Sensitivity Analysis and Publication Bias

The sensitivity analyses conducted across all models demonstrated a high level of consistency, indicating that no single study disproportionately influenced the overall findings.

-For OS based on univariate HRs, the pooled cHR ranged from 1.52 to 1.61 when individual studies were excluded. Despite these minor fluctuations, heterogeneity remained substantial (I^2^ > 82%). No study was identified as having a strong influence on the overall meta-analysis (Appendix A).-In the multivariate analysis of OS, the pooled aHR varied between 1.66 and 1.71 when studies were omitted one by one, with heterogeneity consistently high (I^2^ > 80%). As with the univariate analysis, no individual study significantly affected the results. Full data can be found in Appendix A.-For PFS based on univariate HRs, excluding individual studies led to non-significant changes in the pooled cHR estimate, which ranged from 1.37 to 1.44. Heterogeneity remained considerable (I^2^ > 72%) throughout the analyses, and there was no evidence that any single study dominated the overall results. Further details are available in Appendix A.-In the multivariate analysis of PFS, the exclusion of individual studies caused only minor variations in the pooled HR, ranging from 1.51 to 1.61. Heterogeneity was moderate to high (I^2^ between 64% and 78%) across all iterations. The exclusion of the study by Sugimoto et al. (2018) [61] led to a significant decrease in heterogeneity (I^2^ decreased from 67% to 50%), indicating its substantial contribution to the variability observed in the reported outcome. More detailed results are provided in Appendix A.

Regarding publication bias, Egger’s test was used to assess potential funnel plot asymmetry in all analyses.

-In the OS univariate analysis, the funnel plot revealed several studies outside the expected triangular region, suggesting considerable heterogeneity. The Egger’s test result was highly significant (t = 6.60, *p* < 0.0001), indicating potential asymmetry and suggesting the likelihood of publication bias (Appendix A).-Similarly, in the OS multivariate analysis, the funnel plot showed some studies lying outside the triangular area, consistent with significant heterogeneity. The Egger’s test indicated strong evidence of asymmetry (t = 7.24, *p* < 0.0001), further supporting the presence of publication bias in this group (Appendix A).-For the PFS univariate analysis, the funnel plot appeared more symmetrical, though a few studies fell outside the expected region. The Egger’s test yielded a borderline result (t = 2.07, *p* = 0.0536), suggesting only a marginal possibility of asymmetry and, consequently, a low likelihood of publication bias in this subgroup (Appendix A).-Finally, in the PFS multivariate analysis, the funnel plot displayed a relatively symmetrical distribution, with studies clustering closely within the triangular region. The Egger’s test was non-significant (t = 0.57, *p* = 0.5765), indicating no strong evidence of asymmetry, and thus, publication bias in this analysis seems unlikely (Appendix A).

## 4. Discussion

This meta-analysis of 48 observational studies aimed to explore the prognostic significance of sarcopenia in patients with PC, focusing on its impact on OS and PFS. Our findings support the growing body of evidence suggesting that sarcopenia is significantly associated with worse clinical outcomes, regardless of the treatment setting or specific sarcopenia measurement method, although some of these factors significantly influence the strength of association.

We found a significant association between sarcopenia and worse OS, both in univariate and multivariate analyses. The pooled cHR for OS based on univariate regression analyses was 1.58 (95% CI, 1.38–1.82), while in multivariate analyses, the aHR was 1.67 (95% CI, 1.47–1.90). These results are consistent with other large-scale studies. For instance, Mintziras et al. (2018) found that sarcopenia was associated with a 49% increase in mortality risk in PC patients, with a cHR of 1.49 (95% CI, 1.27–1.74), a value comparable to our findings [86]. Similarly, Pierobon et al. (2021) identified sarcopenia as a key determinant of worse survival in PC, with a 14% reduction in OS for sarcopenic patients [87].

PC is characterized by a high systemic inflammatory response and cachexia, conditions that promote muscle wasting [88]. In fact, cachexia and sarcopenia are often closely intertwined in this population. Pancreatic tumors secrete pro-inflammatory cytokines such as IL-6 and TNF-α, which promote muscle protein degradation through the ubiquitin–proteasome pathway [86,87,89,90]. Additionally, chemotherapy in PC frequently exacerbates muscle loss due to its toxicity and the accompanying inflammatory response, leading to sarcopenia-induced metabolic stress that worsens patient outcomes [10,91]. Thus, sarcopenia may directly impair the body’s ability to tolerate aggressive treatments like surgery and chemotherapy, increasing mortality risk.

Regarding PFS, our findings also confirm that sarcopenia is associated with worse outcomes. The pooled cHR for PFS based on univariate analyses was 1.39 (95% CI, 1.16–1.67), and it was 1.55 (95% CI, 1.29–1.86) in multivariate analyses. This aligns with the results from Zhong et al. (2024), who also found that sarcopenia is a key predictor of disease progression in various cancers [17]. The consistent association between sarcopenia and worse PFS across studies underscores the importance of early identification and management of sarcopenia to potentially improve cancer treatment outcomes. A biologically plausible hypothesis for this lies in the fact that sarcopenia reduces patients’ resilience in withstanding the cumulative physical stress of cancer therapies. In fact, lower muscle mass and function reduces their physical capacity to recover between chemotherapy cycles, delays treatment schedules, and forces dose reductions—all of which can lead to earlier disease progression. This is the rationale for some oncological strategies aimed at improving body composition parameters to reduce the toxic effects of cancer [92]. Notably, authors like Bundred et al. (2019) suggested that sarcopenic patients with pancreatic (and colorectal) cancer are particularly vulnerable to early disease progression due to their tumors’ aggressive metabolic demands, which are compounded by the already catabolic state induced by sarcopenia [93].

Moreover, the high heterogeneity observed in our analyses is in line with previous research that highlights variability in sarcopenia measurement methods, particularly in how SMI is used to define sarcopenia. Ratnayake et al. (2018) and Thormann et al. (2023) also highlighted this variability, pointing to the need for standardized diagnostic criteria to reduce heterogeneity and improve comparability across studies [94,95]. This is consistent with previous findings from our group [24] and could reflect how the diverse tumor profiles, patient demographics, and treatment approaches across studies influence the prognostic value of sarcopenia. However, it should be noted that the absolute magnitude of the variance denoted by the observed τ^2^ values was low or moderate, suggesting that the practical influence of heterogeneity on the overall effect size is not excessively large.

On the other hand, our subgroup analyses showed interesting results which have not been sufficiently addressed in the currently available literature. We examined the impact of different methods used to define sarcopenia and the SMI cutoff values on survival outcomes. Studies using SMI to define sarcopenia consistently reported stronger associations with both OS and PFS compared to those using other methods. The HR for OS in studies using SMI was 1.71 (95% CI, 1.49–1.97), compared to 1.52 (95% CI, 1.30–1.78) for studies using other methods. This is consistent with the findings of Thormann et al. (2023), who found that SMI-based definitions of sarcopenia provided more consistent prognostic information compared to alternative definitions based on functional assessments [95]. Similarly, Pierobon et al. (2021) explored functional and alternative definitions of sarcopenia, suggesting that these may not consistently capture the prognostic impact as effectively as structural measures like SMI [87].

The choice of SMI cutoff values also influenced the reported associations between sarcopenia and survival. Studies using an SMI cutoff of <50 cm^2^/m^2^ reported stronger associations with OS and PFS than those using a higher cutoff. Pierobon et al. (2021) and Mintziras et al. (2018) argued that lower SMI cutoffs may better capture the severity of muscle wasting and its impact on survival [86,87]. Standardizing SMI cutoff values may therefore improve the consistency of sarcopenia-related survival predictions across studies [24,72].

Another key finding from our subgroup analyses was the differential impact of sarcopenia on survival outcomes depending on the treatment intent (curative vs. palliative). Studies conducted in curative settings reported stronger associations between sarcopenia and OS (HR 1.75, 95% CI, 1.49–2.06) compared to those conducted in palliative settings (HR 1.41, 95% CI, 1.22–1.62). This is consistent with findings from Bundred et al. (2019), who observed that sarcopenia’s impact on survival is more pronounced in patients undergoing potentially curative treatments [93]. In curative settings, patients must endure aggressive treatment regimens, and those with sarcopenia are less likely to tolerate these therapies, leading to higher mortality. Preoperative interventions to address sarcopenia, such as nutrition and resistance exercise, may mitigate these risks and improve outcomes in patients undergoing curative surgery.

The results of this meta-analysis provide further evidence that sarcopenia is a robust predictor of worse survival outcomes in cancer patients, regardless of the variability in the imaging criteria used to define it across studies, or of prevalence or treatment intention. Given the consistent association between sarcopenia and poorer outcomes across studies, incorporating sarcopenia assessments into routine cancer care could help clinicians identify high-risk patients and tailor treatment accordingly [96]. Identifying sarcopenic patients may allow care providers to implement supportive measures or adjust treatment regimens to improve outcomes [97], and strategies to mitigate sarcopenia, such as nutritional interventions, physical therapy, and rehabilitation, should be prioritized, particularly in patients undergoing curative treatments [98].

Despite the robust findings of this meta-analysis, several limitations should be acknowledged. First, the high level of heterogeneity across studies limits the generalizability of our results. Although we employed random-effects models and conducted subgroup analyses to explore potential sources of heterogeneity, residual variability remains. Notably, although the I^2^ values were generally high, the τ^2^ values were mostly low and occasionally moderate (0.01–0.25). Additionally, the use of different sarcopenia measurement methods and cutoff values complicates comparisons between studies, even after stratifying data based on different CT-based indices and thresholds. In fact, our results suggest the presence of misclassification bias among individuals in the included studies due to variability in the cutoffs used to define sarcopenia. However, until a universal consensus is established on CT-based measurements of sarcopenia, it remains challenging to determine the direction and magnitude of this bias. This limitation highlights the urgent need for researchers and international societies to agree on standardized methods and optimal cutoff values for CT-based sarcopenia assessment. Additionally, outcomes derived from multivariate analyses should be interpreted with caution, as the number and type of variables included in each model varied across studies, hindering direct comparisons. However, it is worth noting that most of the evaluated outcomes showed similar crude and adjusted HR values. Finally, publication bias may have influenced our results, particularly in the OS analyses, as suggested by the significant Egger’s test findings. This bias may reflect the tendency for studies reporting significant associations between sarcopenia and survival outcomes to be more likely to be published. Future studies with larger sample sizes and more consistent methodologies are needed to confirm our findings and further elucidate the role of sarcopenia in PC prognosis.

## 5. Conclusions

This meta-analysis provides strong evidence that sarcopenia determined by CT is an independent predictor of worse overall and progression-free survival in pancreatic cancer patients. Our findings highlight the importance of routine sarcopenia assessment and suggest that interventions aimed at mitigating muscle loss could play a key role in improving outcomes in this high-risk population. Future research should focus on standardizing sarcopenia assessment methods and exploring the potential benefits of targeted therapeutic strategies in sarcopenic patients.

## Figures and Tables

**Figure 1 cancers-17-00607-f001:**
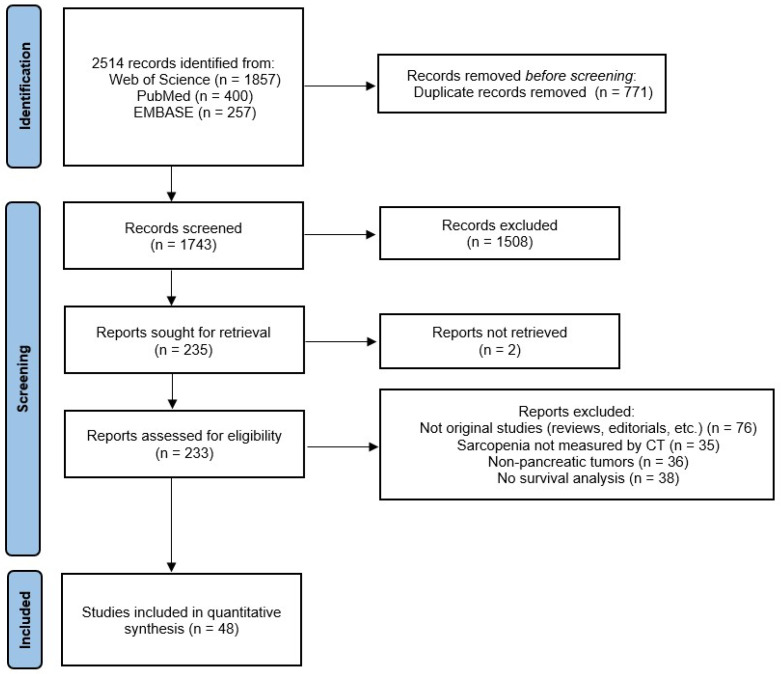
Flow diagram of the study according to the PRISMA guidelines.

**Figure 2 cancers-17-00607-f002:**
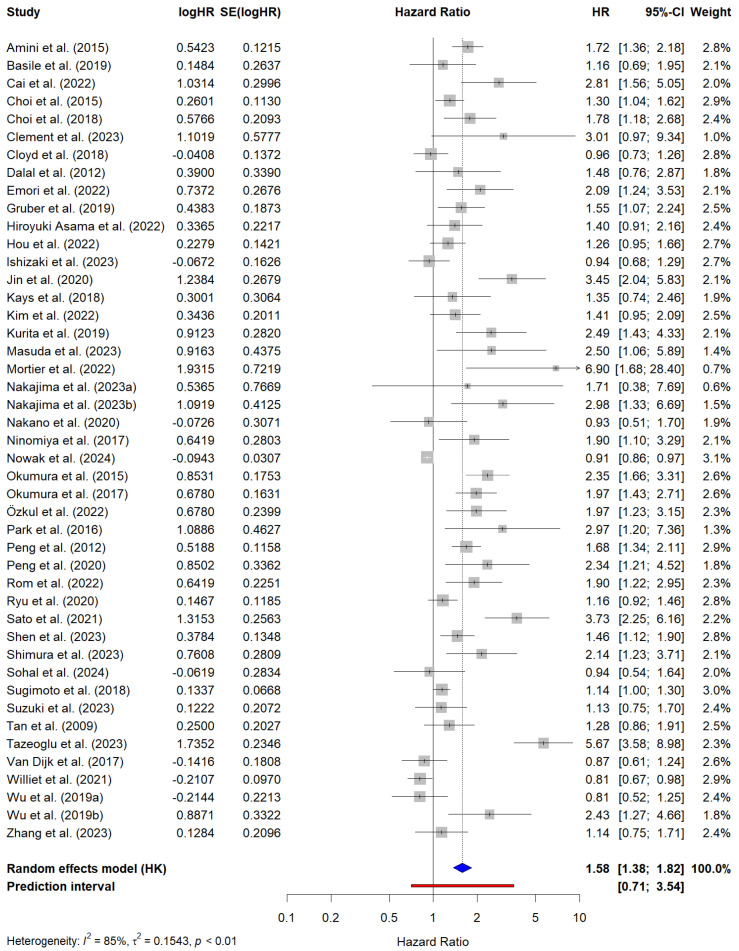
Forest plot of the studies reporting univariate analysis results (i.e., crude hazard ratios -HR-) for the prediction of overall survival in PC patients with sarcopenia. The blue diamond represents the pooled HR and its 95% confidence interval (CI). The red line represents the prediction interval, which provides an estimate of the potential range of HRs in future studies. Amini et al. (2016) [68]; Basile et al. (2019) [69]; Cai et al. (2022) [34]; Choi et al. (2015) [35]; Choi et al. (2017) [36]; Clement et al. (2023) [37]; Cloyd et al. (2018) [38]; Dalal et al. (2012) [39]; Emori et al. (2022) [41]; Gruber et al. (2019) [42]; Hiroyuki Asama et al. (2022) [43]; Hou et al. (2022) [72]; Ishizaki et al. (2023) [44]; Jin et al. (2022) [45]; Kays et al. (2018) [46]; Kim et al. (2022) [47]; Kurita et al. (2019) [49]; Masuda et al. (2023) [23]; Mortier et al. (2022) [50]; Nakajima et al. (2023a) [73]; Najakima et al. (2023b) [73]; Nakano et al. (2021) [51]; Ninomiya et al. (2017) [52]; Nowak et al. (2024) [77]; Okumura et al. (2015) [53]; Okumura et al. (2017) [54]; Özkul et al. (2022) [55]; Park et al. (2016) [76]; Peng et al. (2012) [71]; Peng et al. (2020) [78]; Rom et al. (2022) [7]; Ryu et al. (2020) [56]; Sato et al. (2021) [57]; Shen et al. (2023) [58]; Shimura et al. (2023) [59]; Sohal et al. (2024) [60]; Sugimoto et al. (2018) [61]; Suzuki et al. (2023) [62]; Tan et al. (2009) [63]; Tazeoglu et al. (2023) [64]; Van Dijk et al. (2017) [66]; Williet et al. (2021) [70]; Wu et al. (2019a) [74]; Wu et al. (2019b) [74]; Zhang et al. (2023) [67].

**Figure 3 cancers-17-00607-f003:**
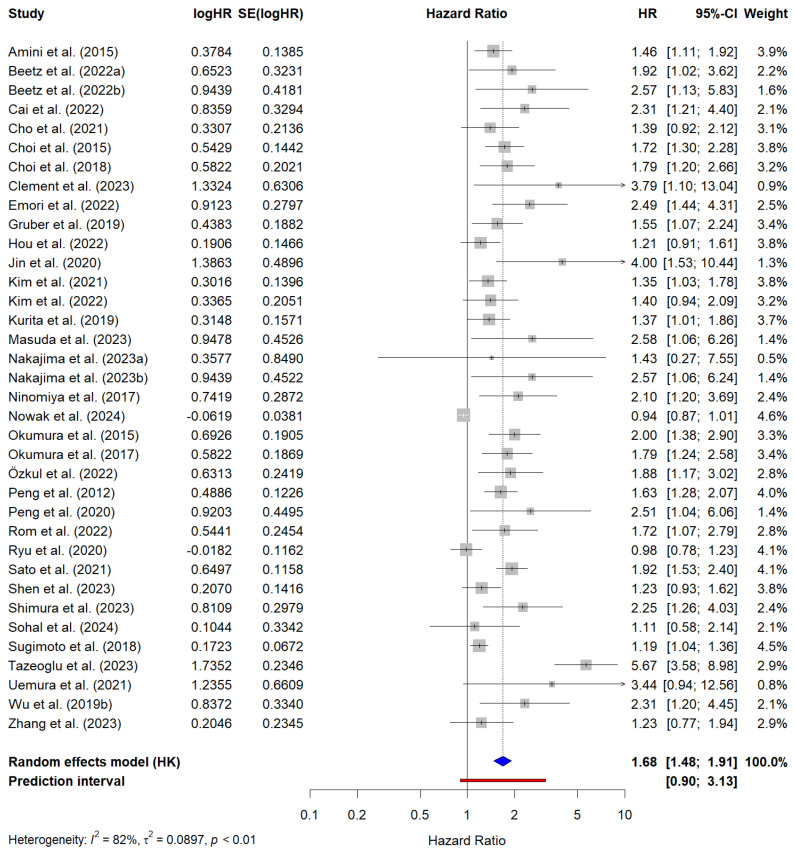
Forest plot of the studies reporting multivariate analysis results (i.e., adjusted hazard ratios) for the prediction of overall survival in PC patients with sarcopenia. The blue diamond represents the pooled HR and its 95% confidence interval (CI). The red line represents the prediction interval, which provides an estimate of the potential range of HRs in future studies. Amini et al. (2016) [68]; Beetz et al. (2022a) [33]; Beetz et al. (2022b) [33]; Cai et al. (2022) [34]; Cho et al. (2021) [75]; Choi et al. (2015) [35]; Choi et al. (2017) [36]; Clement et al. (2023) [37]; Emori et al. (2022) [41]; Gruber et al. (2019) [42]; Hou et al. (2022) [72]; Jin et al. (2022) [45]; Kim et al. (2022) [47]; Kim et al. (2021) [48]; Kurita et al. (2019) [49]; Masuda et al. (2023) [23]; Nakajima et al. (2023a) [73]; Najakima et al. (2023b) [73]; Ninomiya et al. (2017) [52]; Nowak et al. (2024) [77]; Okumura et al. (2015) [53]; Okumura et al. (2017) [54]; Özkul et al. (2022) [55]; Peng et al. (2012) [71]; Peng et al. (2020) [78]; Rom et al. (2022) [7]; Ryu et al. (2020) [56]; Sato et al. (2021) [57]; Shen et al. (2023) [58]; Shimura et al. (2023) [59]; Sohal et al. (2024) [60]; Sugimoto et al. (2018) [61]; Tazeoglu et al. (2023) [64]; Uemura et al. (2020) [65]; Van Dijk et al. (2017) [66]; Williet et al. (2021) [70]; Wu et al. (2019b) [74]; Zhang et al. (2023) [67].

**Figure 4 cancers-17-00607-f004:**
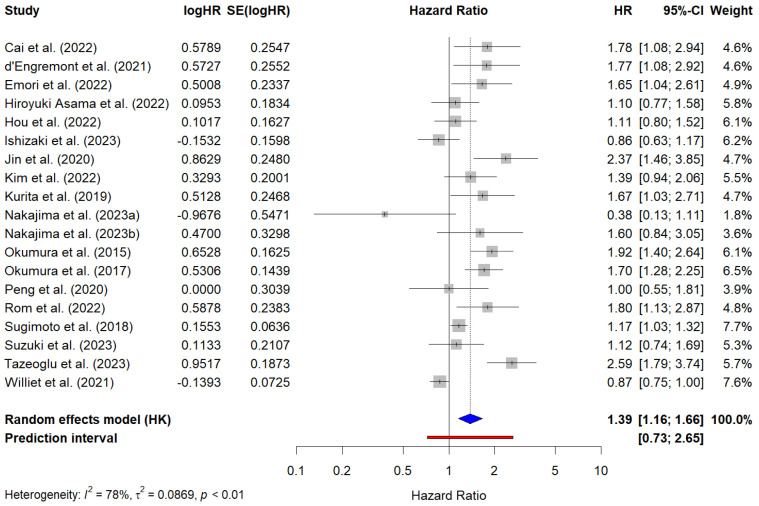
Forest plot of the studies reporting univariate analysis results (i.e., crude hazard ratios) for the prediction of progression-free survival in PC patients with sarcopenia. The blue diamond represents the pooled HR and its 95% confidence interval (CI). The red line represents the prediction interval, which provides an estimate of the potential range of HRs in future studies. Cai et al. (2022) [34]; d’Engremont et al. (2021) [40]; Emori et al. (2022) [41]; Hiroyuki Asama et al. (2022) [43]; Hou et al. (2022) [72]; Ishizaki et al. (2023) [44]; Jin et al. (2022) [45]; Kim et al. (2022) [47]; Kurita et al. (2019) [49]; Nakajima et al. (2023a) [73]; Najakima et al. (2023b) [73]; Okumura et al. (2015) [53]; Okumura et al. (2017) [54]; Özkul et al. (2022) [55]; Park et al. (2016) [76]; Peng et al. (2020) [78]; Rom et al. (2022) [7]; Sugimoto et al. (2018) [61]; Suzuki et al. (2023) [62]; Tazeoglu et al. (2023) [64]; Williet et al. (2021) [70].

**Figure 5 cancers-17-00607-f005:**
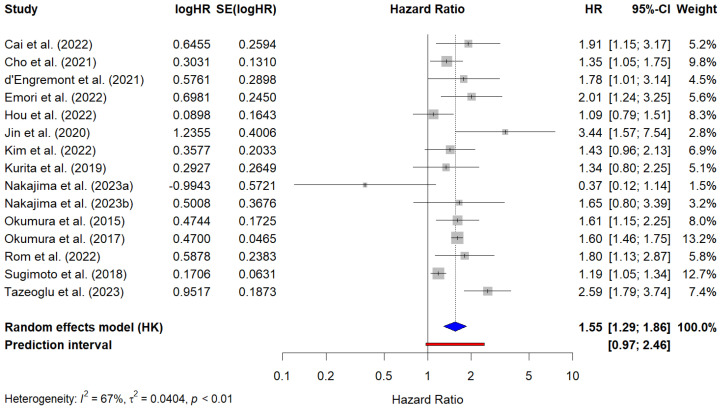
Forest plot of the studies reporting multivariate analysis results (i.e., adjusted hazard ratios) for the prediction of progression-free survival in PC patients with sarcopenia. The blue diamond represents the pooled HR and its 95% confidence interval (CI). The red line represents the prediction interval, which provides an estimate of the potential range of HRs in future studies. Cai et al. (2022) [34]; Cho et al. (2021) [75]; d’Engremont et al. (2021) [40]; Emori et al. (2022) [41]; Hou et al. (2022) [72]; Jin et al. (2022) [45]; Kim et al. (2022) [47]; Kurita et al. (2019) [49]; Nakajima et al. (2023a) [73]; Najakima et al. (2023b) [73]; Okumura et al. (2015) [53]; Okumura et al. (2017) [54]; Rom et al. (2022) [7]; Sugimoto et al. (2018) [61]; Tazeoglu et al. (2023) [64].

**Table 1 cancers-17-00607-t001:** Main characteristics of the studies included in this meta-analysis. ASM, appendicular skeletal muscle. F, female. M, male. m (IQR), median (interquartile range). NOS, Newcastle–Ottawa scale. PC, pancreatic cancer. PDAC, pancreatic ductal adenocarcinoma. PMI, psoas muscle mass index. SMI, skeletal muscle mass index. TPV, total psoas volume. TSM, total skeletal muscle index. X + SD, mean + standard deviation. * From data calculation provided in the methodology of the article, the corresponding values for class I sarcopenia are 57.5 cm^2^/m^2^ and 38.3 cm^2^/m^2^ for men and women, respectively. Further details can be consulted in [24].

Author (Year)	N	Age m (IQR) X ± SD	Women (%)	Sarcopenia (%)	Imaging Index	Definition of Cutoff Value	Sex-Specific Cutoff Values	Tumor Information	Management
Amini et al. (2016) [68]	763	67 (58–74)	45.2	19.9	TPV	Lowest quartile	M < 17.2 cm^2^/m^2^ F < 12.0 cm^2^/m^2^	PDAC	Curative
Basile et al. (2019) [69]	94	45 (48% < 70 years)	44.6	73.4	SMI	Prado et al. [16]	M < 43 cm^2^/m^2^ (BMI < 25); 53 cm^2^/m^2^ (BMI > 25) F < 41 cm^2^/m^2^	Advanced PC	Palliative
Beetz et al. (2022) [33]	103	62 + 11 (37–84)	39.8	63.1	SMI	Prado et al. [16]	M < 52.3 cm^2^/m^2^ F < 38.5 cm^2^/m^2^	PDAC	Not specified
Cai et al. (2022) [34]	115	65.1 + 9	38.2	33	SMI	AUC (best accuracy, outcome: ‘mortality’)	M < 45.16 cm^2^/m^2^ F < 34.65 cm^2^/m^2^	PDAC	Curative
Cho et al. (2021) [75]	299	62 (35–83)	40.4	9.6	SMI	Fujiwara et al. [79]	M < 36.2 cm^2^/m^2^ F < 29.6 cm^2^/m^2^	Locally advanced PC	Palliative
Choi et al. (2015) [35]	484	60.4 (20–85)	39	33.2	SMI	AUC (not specified)	M < 42.2 cm^2^/m^2^ F < 33.9 cm^2^/m^2^	Advanced PC	Palliative
Choi et al. (2018) [36]	180	64.4 + 9.3	45.5	33.3	SMI	Lowest tertile	M < 45.3 cm^2^/m^2^ F < 39.3 cm^2^/m^2^	PC	Curative
Clement et al. (2023) [37]	44	62 (52–68)	52	59	SMI	Prado et al. [16]	M < 43 cm^2^/m^2^ (BMI < 25); <53 (BMI > 25) F < 41 cm^2^/m^2^	Metastatic PC	Palliative
Cloyd et al. (2018) [38]	127	64.6 + 8.9	59	62.9	SKM (=SMI)	Mourtzakis et al. [80]	M < 38.9 cm^2^/m^2^ F < 55.4 cm^2^/m^2^	PDAC	Curative
Dalal et al. (2012) [39]	41	59 (42–81)	56	63.4	SKM (=SMI)	Prado et al. [16]	M < 52.4 cm^2^/m^2^ F < 38.5 cm^2^/m^2^	Locally advanced PC	Palliative
d’Engremont et al. (2021) [40]	98	67.7 (61.8–73.8)	47.8	56.1	SMI	Prado et al. [16]	M < 52.4 cm^2^/m^2^ F < 38.5 cm^2^/m^2^	Localized PDAC	Curative
Emori et al. (2022) [41]	84	<65:30 (36%) >65:54 (64%)	36.9	50	SMI	Nishikawa et al. [81]	M < 42 cm^2^/m^2^ F < 38 cm^2^/m^2^	Unresectable PDAC	Palliative
Gruber et al. (2019) [42]	133	68 (34–87)	48.8	58.6	SMI	Prado et al. [16]	M < 52.4 cm^2^/m^2^ F < 38.5 cm^2^/m^2^	PDAC	Curative
Hiroyuki Asama et al. (2022) [43]	124	69 (40–84)	45.9	50.8	SMI	Nishikawa et al. [81]	M < 42 cm^2^/m^2^ F < 38 cm^2^/m^2^	Unresectable PDAC	Palliative
Hou et al. (2022) [72]	232	<65:139 (59.9) >65 = 93 (40.1)	35.7	49.1	TPA	Prado et al. [16]	M < 545 mm^2^/m^2^ F < 385 mm^2^/m^2^	Advanced PC	Palliative
Ishizaki et al. (2023) [44]	180	<65:90 (50%) >65:90 (50%)	43.8	50	SMI	Nishikawa et al. [81]	M < 42 cm^2^/m^2^ F < 38 cm^2^/m^2^	Unresectable PC	Palliative
Jin et al. (2022) [45]	119	60.2 + 8.4	50.4	47.8	SMI	Nishikawa et al. [81]	M < 41 cm^2^/m^2^ F < 38.5 cm^2^/m^2^	Potentially resectable PDAC	Curative
Kays et al. (2018) [46]	53	59.5 + 9.9	37.7	49	SKMI (=SMI)	Prado et al. [16]	M < 52.4 cm^2^/m^2^ F < 38.5 cm^2^/m^2^	Advanced PC	Palliative
Kim et al. (2022) [47]	347	63.6 + 9.6	41.7	54.1	SMI	Prado et al. [16]	M < 42.2 cm^2^/m^2^ F < 33.9 cm^2^/m^2^	PDAC	Curative
Kim In-Ho et al. (2021) [48]	251	63.4 + 9.4	35.8	40.6	SMI	Outcome-based Contal and O’Quigley method	M < 43 cm^2^/m^2^ (BMI < 25); <53 (BMI > 25) F < 41 cm^2^/m^2^	Metastatic PC	Palliative
Kurita et al. (2019) [49]	82	64 (40–80)	26.8	51.2	SMI	Optimum stratification (log-rank, outcome: ‘mortality’)	M < 45.3 cm^2^/m^2^ F < 37.1 cm^2^/m^2^	PC	Palliative
Masuda et al. (2023) [23]	162	69 (40–85)	44.4	50	SMI	Median value	M < 41.9 cm^2^/m^2^ F < 36.6 cm^2^/m^2^	Localized PDAC	Curative
Mortier et al. (2022) [50]	70	Sarcopenic: 65 (43–85) Non-sarcopenic: 73 (54–80)	52.8	21.4	SMI	Prado et al. [16]	M < 43 cm^2^/m^2^ (BMI < 25); <53 (BMI > 25) F < 41 cm^2^/m^2^	Localized PDAC	Curative
Nakajima et al. (2023)-1 [73]	44	72 (65–76)	61.3	34	TPA	Lowest tertile	M < 7.79 cm^2^/m^2^ F < 5.70 cm^2^/m^2^	Resectable PC	Curative
Najakima et al. (2023)-2 [73]	71	67 (60–72)	59.1	32.3	TPA	Lowest tertile	M < 7.16 cm^2^/m^2^ F < 6.44 cm^2^/m^2^	Borderline resectable PC	Curative
Nakano et al. (2021) [51]	55	67 (35–85)	23.6	49	SMI	Choi et al. [35]	M < 42.2 cm^2^/m^2^ F < 33.9 cm^2^/m^2^	Advanced PC	Palliative
Ninomiya et al. (2017) [52]	265	65.4 + 10.1	38.1	64.1	SMI	Prado et al. [16]	M < 43.75 cm^2^/m^2^ F < 38.5 cm^2^/m^2^	PDAC	Curative
Nowak et al. (2024) [77]	142	64.1 + 10.5	51.4	50.7	SMI	Median value	M < 13.5 cm^2^/m^2^ F < 11.7 cm^2^/m^2^	Advanced PC	Palliative
Okumura et al. (2015) [53]	230	67 (32–87)	46	27.8	PMI	AUC (best accuracy, outcome: ‘death’)	M < 5.9 cm^2^/m^2^ F < 4.1 cm^2^/m^2^	PDAC	Curative
Okumura et al. (2017) [54]	301	68 (61–74)	44.1	39.8	SMI	AUC (best accuracy, outcome: ‘death’)	M < 47.1 cm^2^/m^2^ F < 36.6 cm^2^/m^2^	PC	Curative
Özkul et al. (2022) [55]	115	65.5 + 10.3	29.5	29.5	SMI	AUC (best accuracy, outcome: ‘mortality’)	M < 56.44 cm^2^/m^2^ F < 43.56 cm^2^/m^2^	Unresectable PC	Palliative
Park et al. (2016) [76]	88	65 (34–83)	32.9	86.3	ASM (=SMI)	Conversion from SMI to ASM; <1 SD for young adults	M < 7.50 kg/m^2^ F < 5.38 kg/m^2^ (sarcopenia class I *)	PC	Palliative
Peng et al. (2012) [71]	557	65.7 + 10.6	46.8	24.9	TPA	Lowest quartile	M < 4.92 cm^2^/m^2^ F < 3.62 cm^2^/m^2^	PC	Curative
Peng et al. (2021) [78]	116	66.2 + 11.9	41.3	17.2	SMI	Choi et al. [35]	M < 42.2 cm^2^/m^2^ F < 33.9 cm^2^/m^2^	PC	Curative
Rom et al. (2022) [7]	111	67 (61–75)	46.8	27	SMI	Lowest quartile	M < 44.35 cm^2^/m^2^ F < 34.82 cm^2^/m^2^	PDAC	Curative
Ryu et al. (2020) [56]	548	62.51 (24–88)	40.5	45.9	SMI	Moon et al. [82]	M < 50.18 cm^2^/m^2^ F < 38.63 cm^2^/m^2^	PC (head of pancreas)	Curative
Sato et al. (2021) [57]	112	67.7 (59.2–72.3)	51.7	48.2	SMI	Nishikawa et al. [81]	M < 42 cm^2^/m^2^ F < 38 cm^2^/m^2^	Advanced PDAC	Palliative
Shen et al. (2023) [58]	614	59.9 + 10.3	40	61.5	SMI	Prado et al. [16]	M < 52.4 cm^2^/m^2^ F < 38.5 cm^2^/m^2^	PDAC	Curative
Shimura et al. (2023) [59]	75	67 + 7.8	46.6	60	SMI	AUC	M < 48.4 cm^2^/m^2^ F < 35.5 cm^2^/m^2^	PC	Curative
Sohal et al. (2024) [60]	90	63.2 + 8.5	54.4	35.5	SMI (SMA/BMI)	Not specified (=Prado et al. [16])	M < 52 cm^2^/m^2^ F < 39 cm^2^/m^2^	Resectable PDAC	Curative
Sugimoto et al. (2018) [61]	323	65 (38–88)	45.5	61.9	SMI	Fearon et al. [83] (=Prado et al. [16])	M < 55.4 cm^2^/m^2^ F < 38.9 cm^2^/m^2^	PDAC	Curative
Suzuki et al. (2023) [62]	138	67.5 (59.7–74)	42	44.2	SMI	Nishikawa et al. [81]	M < 42 cm^2^/m^2^ F < 38 cm^2^/m^2^	Unresectable PC	Palliative
Tan et al. (2009) [63]	111	64.4 + 9.3	53.1	55.8	SMI	Prado et al. [16]	M < 59.1 cm^2^/m^2^ F < 48.4 cm^2^/m^2^	PC	Palliative
Tazeoglu et al. (2023) [64]	179	60.45 + 13.08	41.3	46.3	PMI	Bahat et al. [84]	M < 5.3 cm^2^/m^2^ F < 3.6 cm^2^/m^2^	PC	Curative
Uemura et al. (2020) [65]	69	63 (38–74)	44.9	47.8	SMI	Nishikawa et al. [81]	M < 42 cm^2^/m^2^ F < 38 cm^2^/m^2^	Advanced PC	Palliative
Van Dijk et al. (2017) [66]	186	66.5	45.1	33.3	L3-muscle attenuation index (=SMI)	Lowest tertile	M < 45.1 cm^2^/m^2^ F < 36.9 cm^2^/m^2^	PC (head of pancreas)	Curative
Williet et al. (2021) [70]	79	66 (58.5–74)	45.5	69.6	SMI	Optimum stratification (log rank, outcome: ‘mortality’)	M < 55 cm^2^/m^2^ F < 39 cm^2^/m^2^	Metastatic PDAC	Palliative
Wu et al. (2019E) [74]	146	65.5 (36.7–92.2)	63	10.9	TSM (=SMI)	Fujiwara et al. [79]	M < 36.2 cm^2^/m^2^; F < 29.6 cm^2^/m^2^	PC	Not specified
Wu et al. (2019W) [74]	146	65.5 (36.7–92.2)	63	66.4	TSM (=SMI)	Prado et al. [16]	M < 52.4 cm^2^/m^2^ F < 38.5 cm^2^/m^2^	PC	Not specified
Zhang et al. (2023) [67]	113	59 (33–84)	41.5	43.3	SMI	Zeng et al. [85]	M < 44.77 cm^2^/m^2^ F < 32.50 cm^2^/m^2^	PC	Curative

## Data Availability

The data presented in this study are available in this article and Appendix A.

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
