# Peer review of "Computed Tomography-Based Sarcopenia and Pancreatic Cancer Survival—A Comprehensive Meta-Analysis Exploring the Influence of Definition Criteria, Prevalence, and Treatment Intention"

_cancers, 2025, doi:10.3390/cancers17040607_

Round 1

Reviewer 1 Report (Previous Reviewer 1)

Comments and Suggestions for Authors

In the manuscript," Computed tomography-based sarcopenia and pancreatic cancer survival. A comprehensive meta-analysis exploring the influence of definition criteria, prevalence and treatment intention" the authors have highlighted the significance of sarcopenia in pancreatic cancer patients. But it has the following issues:

1. One of the biggest shortcomings is that authors are not showing any new scientific findings in this manuscript, its well established that PDAC patient condition gets worsened due to cachexia and patients with cachexia even survive lesser.

2. It would have been appreciated if they had worked out any new interesting findings or developed a mechanism from their observational studies.

Author Response

Dear reviewer,

Thank you for your comments, which to our knowledge are the same as those provided in the first manuscript review. Therefore, please find below the comments we provided in our previous answer, which in our opinion shall address all your concerns.

We think that the main shortcoming was our (initial) failure to stress the novelty of our contribution rather than its lack of novelty. To understand this, please let us provide you with some context, since we are aware that the topic is complex and there are previous meta-analyses about the association between sarcopenia and PC survival.

This paper is associated with a previous meta-analysis in which we showed that there is a high variability in sarcopenia prevalence based on which criteria are used (SMI, TPA, etc.) and, within the most commonly used criterion (i.e., SMI), based on which threshold was established. In the said paper, we emphasized that one of the main arising problems of such variability lies in the fact that it can lead to inadequate outcome assessments (e.g., mortality and PFS) due to misclassification bias. For example, if the threshold for sarcopenia is set at 54 cm2/m2 for a given patient/population, its prevalence in the sample will be significantly different as compared to setting the threshold at 36 cm2/m2. If such hypothetical study measures patient survival, the results will significantly differ based on the prevalence of sarcopenia. Well, the point is that we have proved that there is a high heterogeneity across previously published studies, which prevents their direct comparisons without considering the potential bias of the criteria used for the definition of sarcopenia. This potential bias has only been controlled in a previous study (Wu et al., included in our meta-analysis).

With this background context, it is easier to understand that the novelty of our work lies in:

-First, we included 48 observational studies. This is, to our knowledge, the meta-analysis with the largest number of studies published on this particular topic to date.

-Second, we performed 16 subgroup analyses contemplating variables from univariate and bivariate analyses. In particular:

            Subgroup analysis based on the method used to estimate sarcopenia

Subgroup analysis based on the cutoff used in SMI

Subgroup analysis based on the prevalence of sarcopenia found in each study

Subgroup analysis based on treatment intention in each study

All these analyses are relevant as they answer to the question: does the variability in the definition of CT-based sarcopenia modify its association with mortality? (and, if so, what do the available data say about such variability?). We found that the overall effect shows an association between sarcopenia and higher OS/PFS, but the cutoff used in SMI and the treatment intention change both the crude and adjusted HRs.

Overall, this provides novel insights on current knowledge about the topic. In our opinion, it adds significant value and reinforces the fact that future studies need to address and control how sarcopenia is measured. However, to emphasize our novel findings, we changed the article title to clarify that we explored how several important confounders were explored in the meta-analysis. Similarly, we made some modifications to the abstract, and we also made some other changes throughout the text. But, in general, we think that the paper offers relevant knowledge not previously addressed on the topic, apart from providing an updated, comprehensive data synthesis.

Reviewer 2 Report (Previous Reviewer 2)

Comments and Suggestions for Authors

I do confirm my previous positive opinion about this manuscript

Author Response

Thank you very much for your positive opinion, which we value a lot. In this reviewed version of our manuscript, we have made some minor changes in the limitations section as suggested by another reviewer.

Reviewer 3 Report (Previous Reviewer 3)

Comments and Suggestions for Authors

In this article, Ramos-Bossini and colleagues submitted a revised version of previously submitted systematic review and meta-analysis (cancers-3265848) evaluating the prognostic role of sarcopenia in pancreatic cancer. As mentioned in the review of the original article, this is not a novel concept and has been demonstrated in other SR/MAs that are cited in this publication as well. This paper however adds some new subgroup analyses.in an attempt to minimize bias and increase the accuracy of the results.

The authors have made improvement compared to the initial version of their article - adding a table of the baseline characteristics of the included studies, further clarified their exclusion criteria and used the tau square statistic to assess statistical heterogeneity, as per the previous recommendations.

The fact that studies are using different cutoffs to define sarcooenia remains problematic and needs to be further emphasized in the limitations - specifically the possibility of misclassification bias occuring.

Author Response

Dear reviewer,

Thank you very much for your comments. We are grateful for the previously provided suggestions and glad to know that they seem satisfactory to you.

Regarding your last comment, we do agree with you and, indeed, our previous meta-analysis on the topic intended to highlight the problem of variability in cutoff definitions for sarcopenia. Although our research was useful to go in depth with this issue, it remains problematic as you point out. In our new manuscript version we have added a new paragraph further emphasizing that our findings suggest that misclassification bias is occurring, but as long as no clear definition on the gold standard is universally established, the direction and the magnitude of such bias is difficult to ascertain.

We hope that the changes made are satisfactory to you, and we remain at your disposal for any further question.

This manuscript is a resubmission of an earlier submission. The following is a list of the peer review reports and author responses from that submission.

Round 1

Reviewer 1 Report

Comments and Suggestions for Authors

In the manuscript,"Sarcopenia is associated with worse overall and progression free survival in pancreatic cancer. A meta-analysis of 48 observational studies" the authors have highlighted the significance of sarcopenia in pancreatic cancer patients. But it has the following issues:

1. One of the biggest shortcomings is that authors are not showing anything new in their manuscript, its well established that PDAC patient condition gets worsened due to cachexia and patients with cachexia even survive lesser. 

2. It would have been appreciated if they have worked out any new interesting finding or develop a mechanism from their observational studies.

Reviewer 2 Report

Comments and Suggestions for Authors

From a biostats and clinical epidemioloy point of view, this SR/MA has been very well planned and realized, and it's a good complement to the previous manuscript coming from the same Spanish research group.

- line 133, why have you not registered into PROSPERO (https://www.crd.york.ac.uk/prospero/)!? it would be useful

- line 186, please verify ref 27, is it correct!?

- line 192, have you performed cumulative MA as sensitivy analysis too? Are results stable over time in your opinion?

- line 199, a table summarizing the 48 studies main characteristics would be of help for the reader

- figures 2 and 3, very good agreement between uni vs multivariable results, of note!

- the same for figures 4 and 5, stable metrics and reliable estimates!

Reviewer 3 Report

Comments and Suggestions for Authors

In this article, Ramos-Bossini and colleagues conducted a systematic review and meta-analysis evaluating the prognostic role of sarcopenia in pancreatic cancer. Admittedly, this does not represent a novel topic and there are similar publications in the literature addressing this topic (examples PMID 36564257, 34300199, 31266698, 30266663) with similar results.

There are significant issues with the overall methodology as well as the presentation of the results in this study:
1) The eligibility criteria are very non-specific and hence the patient population includes patients of any stage, prior treatment (surgical or medical) and treatment intent. 
2) There is not a single table included in the manuscript that describes the individual study characteristics (year, country, population included, cutoffs used, method used to assess sarcopenia) or patient baseline characteristics (age, gender, race, stage, prior surgical treatment, prior medical treatment, other comorbities - and any other factor that could potentially be a confounder).
3) Studies with different cutoffs for a continuous outcome will result in significant misclassification bias, two patients with identical
characteristics could be distributed in the sarcopenic or non-sarcopenic group, depending on the study. This is a fundamental flaw and can lead to biased effect estimates.
4) Pooling adjusted estimates from multivariable models that do not adjust for the same confounding factors is also problematic and does not lead to interpretable pooled result, since the adjustment is not uniform across all pooled studies.
5) Statistical heterogeneity assessment based on I2 is problematic, especially when so many studies are pooled together (PMID 19036172). Tau would be the more appropriate measure here.

Due to the methodological shortcomings in the design of this study, I am unfortunately unable to recommend it for publication.